# Novel Coronavirus Infection (COVID-19) Related Thrombotic and Bleeding Complications in Critically Ill Patients: Experience from an Academic Medical Center

**DOI:** 10.3390/jcm10235652

**Published:** 2021-11-30

**Authors:** Thejus Jayakrishnan, Aaron Haag, Shane Mealy, Corbyn Minich, Abraham Attah, Michael Turk, Nada Alrifai, Laith Alhuneafat, Fadi Khoury, Adeel Nasrullah, Patrick Wedgeworth, Melissa Mosley, Kirtivardan Vashistha, Veli Bakalov, Abhishek Chaturvedi, Swathi Sangli

**Affiliations:** 1Department of Hematology and Medical Oncology, Cleveland Clinic, Cleveland, OH 44104, USA; thejus128@gmail.com; 2Department of Internal Medicine, Allegheny Health Network, Pittsburgh, PA 15222, USA; aaron.haag@ahn.org (A.H.); shane.mealy@ahn.org (S.M.); corbyn.minich2@ahn.org (C.M.); attah.abraham@ahn.org (A.A.); michael.turk@ahn.org (M.T.); nada.alrifai93@gmail.com (N.A.); laith.alhuneafat@ahn.org (L.A.); fadi.khoury93@gmail.com (F.K.); adeel.nasrullah@ahn.org (A.N.); Melissa.Mosley@ahn.org (M.M.); Kirtivardan.Vashistha@ahn.org (K.V.); Veli.Bakalov@ahn.org (V.B.); 3Department of Bioinformatics, University of Washington, Seattle, WA 98195, USA; james.wedgeworth@gmail.com; 4Department of Cardiology, Virginia Commonwealth University, Richmond, VA 23284, USA; abhishek.chaturvedi2391@gmail.com; 5Department of Pulmonary and Critical Care Medicine, Allegheny Health Network, Pittsburgh, PA 15222, USA

**Keywords:** COVID19, SARS-CoV-2, ICU, thrombosis, bleeding mortality

## Abstract

Introduction: Thrombosis and bleeding are recognized complications of the novel coronavirus infection (COVID-19), with a higher incidence described particularly in the critically ill. Methods: A retrospective review of COVID-19 patients admitted to our intensive care units (ICU) between 1 January 2020 and 31 December 2020 was performed. Primary outcomes included clinically significant thrombotic and bleeding events (according to the ISTH definition) in the ICU. Secondary outcomes included mortality vis-a-vis the type of anticoagulation. Results: The cohort included 144 consecutive COVID-19 patients with a median age of 64 years (IQR 54.5–75). The majority were male (85 (59.0%)) and Caucasian (90 (62.5%)) with a median BMI of 30.5 kg/m^2^ (IQR 25.7–36.1). The median APACHE score at admission to the ICU was 12.5 (IQR 9.5–22). The coagulation parameters at admission were a d-dimer level of 109.2 mg/mL, a platelet count of 217.5 k/mcl, and an INR of 1.4. The anticoagulation strategy at admission included prophylactic anticoagulation for 97 (67.4%) patients and therapeutic anticoagulation for 35 (24.3%) patients, while 12 (8.3%) patients received no anticoagulation. A total of 29 patients (20.1%) suffered from thrombotic or major bleeding complications. These included 17 thrombus events (11.8%)—8 while on prophylactic anticoagulation (7 regular dose and 1 intermediate dose) and 9 while on therapeutic anticoagulation (*p*-value = 0.02)—and 19 major bleeding events (13.2%) (4 on no anticoagulation, 7 on prophylactic (6 regular dose and 1 intermediate dose), and 8 on therapeutic anticoagulation (*p*-value = 0.02)). A higher thrombosis risk among patients who received remdesivir (18.8% vs. 5.3% (*p*-value = 0.01)) and convalescent serum (17.3% vs. 5.8% (*p*-value = 0.03%)) was noted, but no association with baseline characteristics (age, sex, race, comorbidity), coagulation parameters, or treatments (steroids, mechanical ventilation) could be identified. There were 10 pulmonary embolism cases (6.9%). A total of 99 (68.8%) patients were intubated, and 66 patients (45.8%) died. Mortality was higher, but not statistically significant, in patients with thrombotic or bleeding complications—58.6% vs. 42.6% (*p*-value = 0.12)—and higher in the bleeding (21.2%) vs. thrombus group (12.1%), *p*-value = 0.06. It did not significantly differ according to the type of anticoagulation used or the coagulation parameters. Conclusions: This study describes a high incidence of thrombotic and bleeding complications among critically ill COVID-19 patients. The findings of thrombotic events in patients on anticoagulation and major bleeding events in patients on no or prophylactic anticoagulation pose a challenging clinical dilemma in the issue of anticoagulation for COVID-19 patients. The questions raised by this study and previous literature on this subject demonstrate that the role of anticoagulation in COVID-19 patients is worthy of further investigation.

## 1. Introduction

Since its emergence, the novel coronavirus infection (COVID-19) has remained a global health crisis and continues to impose significant social, psychological, and logistical burdens on individuals and health care systems [1]. Robust research has resulted in a better understanding of the natural history of the disease, the characteristics of the patients, and the predictors of outcomes. Studies suggest that up to 6% of COVID-19 patients in the US require ICU admission and experience the highest rate of mortality [2]. COVID-19-associated thrombotic and bleeding complications are increasingly recognized as complications of the novel coronavirus (COVID-19), with a higher incidence described particularly in the critically ill. This is described to occur from a cascade of immune responses, which lead to hyperinflammation, endotheliopathy, deranged hemostasis, and fibrinolysis, resulting in hypercoagulability and severe thromboembolic complications [3,4]. These patients exhibit a spectrum of microvascular and macrovascular, arterial, and venous thrombotic complications, including deep vein thrombosis, pulmonary embolism, arterial thrombosis, myocardial infarction, and even large-vessel ischemic strokes. Bleeding is less commonly reported compared to thrombosis [5,6].

The incidence of venous thromboembolism upon coronavirus infection significantly differs depending on the variable ultrasound screening practices, ranging from as low as 2% to as high as 69%. Regardless, the incidence of thrombotic complications is higher in patients admitted to intensive care units (ICU) [7,8,9,10,11,12,13,14,15]. However, COVID-19-associated thrombotic complications are highly associated with a worse prognosis [16]. The current guidelines are primarily based on the interim results of three major clinical trials [17,18,19,20]. Further studies would help us to determine the definitive impact, safety, and efficacy of the use of intermediate- compared to full-dose therapeutic anticoagulation to prevent thrombotic complications. With the potential bleeding risk in the critically ill and in the absence of established indications for anticoagulation, determining the effective anticoagulation dose is complex.

Allegheny Health Network (AHN) is one of the big consortiums of academic (urban) and non-academic (rural) hospitals in Western Pennsylvania. It has been at the forefront of the battle against COVID-19 in Pennsylvania [21,22,23]. Here, we have sought to describe COVID-19-associated thrombotic and major bleeding in patients admitted to our ICU and assess the associated risk factors that would help in the risk stratification of patients and assist in determining our anticoagulation strategies going forward.

## 2. Methods

### 2.1. Study Population

We conducted a retrospective study of all COVID-19 patients consecutively admitted to the AHN’s intensive care units (ICU) between 1 January 2020 and 31 December 2020. All the patients were diagnosed with COVID-19 by real-time polymerase chain reaction (RT-PCR) tests via nasal swabs. The cohort included patients transferred to ICU for the escalation of care and direct admissions from the emergency department. For patients with readmissions, only the index hospitalization due to COVID-19 is included in this study. The independent clinical judgment of providers determined the patients’ disposition to ICU. The severity of illness was determined by the sequential organ failure assessment (SOFA) and the acute physiology and chronic health evaluation 2 (APACHE 2) scoring system. The study excluded patients previously on therapeutic anticoagulation due to arrhythmias or prior history of thrombus.

The AHN institutional review board committee approved the study and waived the need for informed consent. Deidentified data was collected from the AHN electronic health record system (Epic) and included patient demographics, medical history, home medications, clinical presentation characteristics, and treatment strategies. A Charlson Comorbidity Index (CCI) was also calculated for analysis.

### 2.2. Outcomes Analyzed Were as Follows

#### 2.2.1. Primary Outcomes

Clinically significant thrombosis of the different types, including DVT, hepatic vein thrombosis, and arterial thrombosis.Clinically significant bleeding of the different types as defined by the ISTH definition of having a symptomatic presentation and (1) fatal bleeding and/or (2) bleeding in a critical area or organ (such as intracranial, intraspinal, intraocular, retroperitoneal, intra-articular, pericardial, or intramuscular bleeding with compartment syndrome) and/or (3) bleeding causing a fall in hemoglobin level to 20 g L^−1^ (1.24 mmol L^−1^) or less or leading to the transfusion of two or more units of whole blood or red cells [24].

#### 2.2.2. Secondary Outcomes

Mortality.Correlative analysis of development of thrombus with the type of anticoagulation.

Our academic medical center, comprising a multidisciplinary team, developed a treatment algorithm based on a review of several early published studies at the beginning of the pandemic. Please refer to Figure 1 for our treatment algorithm and dosing strategies. We utilized a physician-based risk stratification of a patient’s bleeding risk on admission, which included various coagulation parameters and imaging studies. Prophylactic anticoagulation was indicated for all patient populations with low bleeding risk and low-risk COVID-19 disease, unless the patients had a contraindication, were previously on therapeutic anticoagulation for a specific etiology, or had a history of heparin-induced thrombocytopenia. In contrast, those with high-risk COVID-19 disease requiring higher respiratory support (>6 L NC) were recommended to undergo baseline imaging to assess them for thrombosis. However, some of the patients enrolled during the early periods of the study did not undergo baseline imaging. Patients with evidence of thrombosis were treated with therapeutic anticoagulation, and those without VTE were treated with an intermediate-dosed anticoagulation strategy. Since many patients transitioned between anticoagulants depending on their clinical status, the anticoagulant regimen at admission to the ICU and during the thrombotic or bleeding event were used in the analysis.

### 2.3. Statistical Analysis

Categorical outcomes are described in percentages and continuous variables with medians and interquartile ranges (IQR). Group comparisons of proportions were performed using the χ^2^ test, while continuous variables were compared using the multiple analysis of variance (MANOVA) test or the Student’s *t*-test as appropriate. Univariate logistic regression analysis for mortality was performed for variables with at least 20 observations. Multivariate regression was not performed due to the small sample size and established interactions amongst different variables resulting in a model that may not add meaningful additional information to the existing results. Statistical tests were 2-tailed and statistical significance was defined as a *p*-value ≤ 0.05. Analyses were performed using Stata version 15.1 (Stata Corp., College Station, TX, USA). Since the analyses were not adjusted for multiple comparisons and were prone to type-I error, the findings should be interpreted as exploratory.

## 3. Results

### 3.1. Characteristics of Patients Admitted to ICU

The cohort included 144 consecutive COVID-19 patients with a median age of 64 years (IQR 54.5–75). The characteristics are outlined in Table 1. The majority were male (85 (59.0%)) and Caucasian (90 (62.5%)) with a median body mass index (BMI) of 30.5 kg/m^2^ (IQR 25.7–36.1). The median APACHE score at admission to the ICU was 12.5 (IQR 9.5–22) and the median CCI score was 1 (IQR 0–3). The coagulation parameters at admission to the ICU were a d-dimer level of 1.5 mg/mL (IQR 1.1–4.7), a platelet count of 217.5 k/mcl (IQR 150–288), and an international normalized ratio (INR) of 1.4 (IQR 1.1–1.3).

### 3.2. ICU Treatments

Following admission to the ICU, 99 (68.8%) patients required mechanical ventilation, 80 (55.6%) patients required vasopressor support, 20 (13.8%) patients needed renal replacement therapy, and 7 (4.9%) patients needed mechanical circulatory support. Pharmacological treatment for COVID-19 and respiratory failure management included steroids in 46 (31.9%) patients, remdesivir in 69 (47.9%) patients, neuromuscular paralysis in 49 (35.0%) patients, and convalescent plasma in 75 (52.1%) patients.

### 3.3. Thrombotic or Bleeding Complications

The anticoagulation strategy at admission included prophylactic anticoagulation for 97 (67.4%) patients and therapeutic anticoagulation for 35 (24.3%) patients, while 12 (8.3%) patients received no anticoagulation. The reasons for anticoagulation may have been high suspicion of a DVT/PE by the treatment team prior to transfer to the ICU or new arrhythmias. The exact reason for the anticoagulation prior to transfer to the ICU, and its timing, was not collected during data extraction. A total of 29 (20.1%) patients suffered from thrombus or major bleeding. These included 17 new thrombus events (11.8%) (8 while on prophylactic anticoagulation (7 regular dose and 1 intermediate dose) vs. 9 on therapeutic anticoagulation (*p*-value = 0.02)) and 19 major bleeding (13.2%)—4 on no anticoagulation vs. 7 on prophylactic (6 regular dose and 1 intermediate dose) vs. 8 on therapeutic anticoagulation (*p*-value = 0.02). There were patients who had both thrombotic and bleeding complications. There were 10 pulmonary embolism cases (6.9%). The outcomes are summarized in Table 2.

Higher thrombosis risk among patients who received remdesivir (18.8% vs. 5.3% (*p*-value = 0.01)) and convalescent serum (17.3% vs. 5.8% (*p*-value = 0.03)) were noted. No other associations were noted, including baseline characteristics—gender (male 14% vs. female 8.5%, *p*-value = 0.3), age (*p*-value = 0.88), different races (*p*-value = 0.22), comorbidity index (*p*-value = 0.52), APACHE score (*p*-value = 0.79), BMI (*p*-value = 0.72); coagulation parameters—d-dimer (*p*-value = 0.58), platelets (*p*-value = 0.22), INR (*p*-value = 0.06); ICU treatments—steroid use vs. none (10.9% vs. 12.2%, *p*-value = 0.81), mechanical ventilation vs. none (14.1% vs. 6.7%, *p*-value = 0.20), mechanical circulatory support vs. none (28.6% vs. 11.0%, *p*-value = 0.16), neuromuscular blockade vs. none (16.3% vs. 9.9%, *p*-value = 0.27), or the use of pressors vs. none (16.3% vs. 6.3%, *p*-value = 0.07).

### 3.4. ICU Complications and Mortality

Of the entire cohort, 66 patients (45.8%) died. Mortality was higher, but not statistically significant, in patients with thrombotic or bleeding complications (58.6% vs. 42.6% (*p*-value = 0.12)) and higher in the bleeding (21.2%) vs. the thrombus group (12.1%), *p*-value = 0.06. It did not significantly differ by the type of anticoagulation (*p*-value = 0.57) or according to the coagulation parameters.

## 4. Discussion

Our study suggests that COVID-19 among critically ill patients is associated with a high incidence of complications from the derangement of the coagulation system, and this pertains to both thrombotic and bleeding complications.

While the underlying mechanisms are yet to be clearly elucidated, the current literature elaborating its pathophysiology suggests that the viral infection, with its uninhibited replication, results in endothelial dysfunction and consequently systemic hyperinflammation. The latter is believed to precipitate thrombin generation, high levels of d-dimer, fibrin degradation products, thrombocytopenia, and prolonged clotting times [25]. Studies using viscoelastic methods on these patients have also demonstrated hypercoagulability and fibrinolysis shutdown even for thromboprophylaxis that was deemed appropriate [4]. Furthermore, endothelial dysfunction is known to cause the characteristic procoagulant response with frequently noted micro and macrothrombi, which correlates directly with fibrin elevation and, in severe cases, the elevation of inflammatory cytokines, including tumor necrosis factor, IL-1, and IL-6 [25]. The presence of some of these markers by itself is associated with poor COVID-19 outcomes, possibly with a contribution from the dysfunction of the coagulation system [6,26].

Our study noted a high rate of thrombotic events in critically ill patients, which is consistent with most of the reported evidence thus far [6,27,28,29]. However, in a recent multicenter cohort study in the US among COVID-19 patients [5], the incidence of thrombotic events in COVID-19 patients was found to be lower than previous evidence and, indeed, the findings of our study suggest. Notably, our patients developed thrombotic events despite being treated with prophylactic or therapeutic anticoagulation. A higher thrombosis risk was also seen among patients receiving remdesivir and convalescent plasma, an association that has not been reported previously [30,31]. This should be interpreted within the limitations of the present study and evaluated in future studies.

On the other end of the spectrum, initial studies demonstrated severe bleeding events at an increased rate among hospitalized patients on therapeutic anticoagulation compared to patients receiving prophylactic AC or no AC [9,32]. Our study noted a higher bleeding incidence than in the recently reported literature, and half of those with major bleeding events were on no anticoagulation or only on the thrombo-prophylactic dose [5,18]. The timing of the occurrence of these complications during the patient’s hospitalization has been previously reported to have a biphasic distribution [33]. Major bleeding events occurred later in the hospital stay and followed a decrease in fibrinogen and d-dimer several days prior, although we could not corroborate this distribution [33]. This high bleeding incidence further reinforces the complicated nature of the effect of COVID-19 infection on the coagulation system, regardless of the anticoagulation dose, and underlines the need for additional studies to identify risk factors for patients who succumb to these complications.

The spectrum of coagulation-system dysfunction associated with COVID-19 infection raises several questions about the timing and efficacy of anticoagulation strategies and the need to identify different phenotypes to enable risk stratification. Indeed, several studies have examined the effect of empiric anticoagulation (AC) on hospitalized COVID-19 patients. Based on the initial limited observational studies reported, the American Society of Hematology recommended the use of thromboprophylactic-dose rather than intermediate- or therapeutic-dose anticoagulation in ICU patients and other hospitalized patients [5,9,17,32]. However, other investigators have demonstrated the benefits of the initiation of an early therapeutic dose of AC compared with no AC, with a 27% decrease observed in 30-day mortality without any increased risk of severe bleeding events [34]. This was contradicted by a large open-label multiplatform controlled trial that studied the use of therapeutic anticoagulation as an initial strategy in critically ill patients and demonstrated no survival benefit amongst the therapeutic AC group in comparison with the thromboprophylaxis group [35]. Interestingly, the same study demonstrated the survival benefit of hospital discharge along with decreased requirements for cardiorespiratory support among noncritically ill patients who were put on therapeutic anticoagulation compared to the thromboprophylaxis group [36]. The postulated reason behind this benefit appears to be that the severely ill patients in the ICU may have significantly advanced derangements in coagulation that are no longer amenable to antithrombin inhibition with heparin, while moderately ill patients might have derangements that are less resistant and still amenable to heparin [37]. With the current level of evidence, it is clear that the debate about the appropriate anticoagulation strategy is likely to continue despite this multiplatform trial, especially given the limitations noted in the study.

The limitations of our study include those inherent to the retrospective study design, such as recall biases, indication biases, and the presence of uncharacterized confounding factors. These limitations can be overcome only through prospective cohort studies, which we recommend. While a multivariate regression analysis could have helped us to identify how various factors impacted mortality in a combined model, it was not performed due to the complexity of the resultant model, which failed to add to the presented results significantly. Concerns may also include the potential lack of power and small number of events, which were factors beyond our control given the predefined eligibility criteria. We plan to revisit the results in the future with a larger, updated dataset to address some of these limitations. The follow-up was limited to the inpatient hospital stay and did not capture complications or readmission post-discharge. The data included are from two different hospitals, and variations may exist due to the practice and population patterns of the individual hospitals. Additionally, the use of viscoelastic methods to identify patients at risk for coagulopathy is also being investigated [4].

## 5. Conclusions

This study describes a high incidence of thrombotic and bleeding complications among critically ill COVID-19 patients. The findings of thrombotic events in patients on anticoagulation and major bleeding events in patients on no or prophylactic anticoagulation pose a challenging clinical dilemma in the issue of anticoagulation for COVID-19 patients. The questions raised by this study and previous research on the subject demonstrate that the role of anticoagulation in COVID-19 patients is worthy of further investigation.

## Figures and Tables

**Figure 1 jcm-10-05652-f001:**
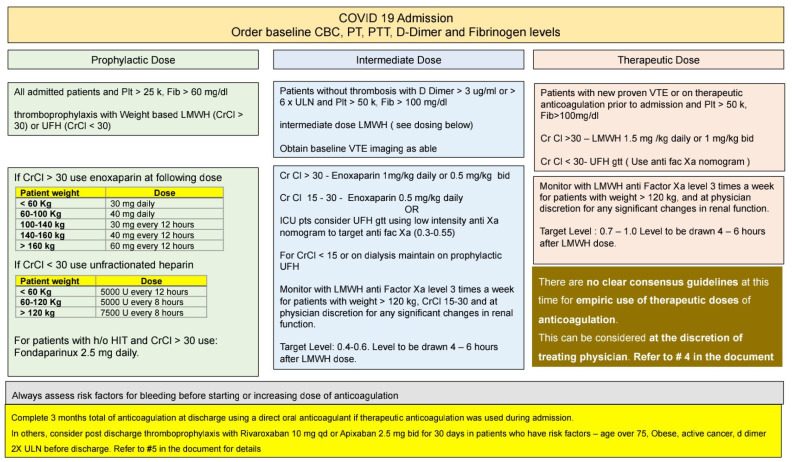
Institutional COVID-19 anticoagulation guideline at Allegheny Health Network.

**Table 1 jcm-10-05652-t001:** Characteristics of patients admitted to the ICU with COVID-19.

Characteristics (*n* = 144)	Percentage or Median (IQR)
Age (in years)	64 (54.5–75)
Gender	
Male	85 (59.0%)
Female	59 (41.0)
Race	
Non-Hispanic White	90 (62.5%)
Non-Hispanic Black	36 (25.0%)
Others	18 (12.5%)
Comorbidities	
Cardiovascular ^a^	102 (70.8%)
Obesity (BMI > 30 kg/m^2^)	76 (52.8%)
Diabetes	57 (39.6%)
Chronic obstructive pulmonary disease	18 (12.5%)
Renal ^b^	24 (16.7%)
CCI	1 (0–3)
Coagulation Labs at Transfer	
d-dimer	1.5 mg/mL (1.1–4.7)
Platelets	217.5 k/mcl (150–288)
International normalized ratio (INR)	1.4 (1.1–1.3)
ICU Admission Severity of Illness	
APACHE-2 (40)	12.5 (9.5–22)
Treatment Strategies	
Mechanical ventilation	99 (68.8%)
Pressors	80 (55.6%)
Neuromuscular blockade use	49 (35.0%)
Steroids	46 (31.9%)
Renal replacement therapy	20 (13.8%)
Mechanical circulatory support	7 (4.9%)
Convalascent plasma	75 (52.1%)
Remdesivir	69 (47.9%)

IQR = interquartile range, BMI = body mass index, CCI = Charlson Comorbidity Index, APACHE = acute physiology and chronic health evaluation. ^a^ Documented history of coronary artery disease or hypertension. ^b^ Documented history of chronic kidney disease.

**Table 2 jcm-10-05652-t002:** Outcomes of patients admitted to ICU stratified by the type of anticoagulation.

	Prophylactic Anticoagulation	Intermediate Dose Anticoagulation	Therapeutic Anticoagulation	None	Total
Number of patients	89 (61.8%)	8 (5.6%)	35 (24.3%)	12 (8.3%)	144
Thrombotic Complications	7	1	9	0	17 (11.8%)
Major Bleeding Complications	6	1	8	4	19 (13.2%)
Death	36 (54.6%)	6 (9.1%)	17 (25.8%)	7 (12%)	66 (45.8%)

## Data Availability

All authors had full access to all the data in the study. We take full responsibility for the integrity of the data and the accuracy of the analysis as well as sharing the data with any interested investigators.

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
