# Peer review of "Novel Coronavirus Infection (COVID-19) Related Thrombotic and Bleeding Complications in Critically Ill Patients: Experience from an Academic Medical Center"

_jcm, 2021, doi:10.3390/jcm10235652_

Round 1

Reviewer 1 Report

The issue addressed by the authors is of paramount importance. However, a major limitation is the lack of power, the very small number of events, the heterogeneity of the groups (several subgroups) limits the validity of the authors' conclusions. 

Author Response

Thank you for reviewing our article. Kindly note the response below

Reviewer 1:

The issue addressed by the authors is of paramount importance. However, a major limitation is the lack of power, the very small number of events, the heterogeneity of the groups (several subgroups) limits the validity of the authors' conclusions. 

Response: We thank the reviewer for their comments. These are valid limitations that we have acknowledged in the discussion. We have updated the limitations to make sure the reviewer’s comments are included. Several of these factors were beyond our control (the sample size, the number of events). The present study is descriptive with limitations inherent to retrospective analysis and should be interpreted keeping those in mind as the reviewer rightly pointed out. It is interesting that even the randomized trials on this topic has shown contradictory results as mentioned in the discussion. Hence, the need for more better designed studies which is what we recommend. 

Reviewer 2 Report

US authors’ manuscript on the relationship between thrombotic and bleeding complications and severe COVID-19. The manuscript is prepared carefully. However, authors should clarify what is new in their research compared to previous works in this field.

Please find below my comments. I am asking the authors to include them in the next version of the manuscript:

  1. Would you please add the name of the country to your affiliation?
  2. In the introduction, the authors should also refer to the disorders of fibrinolysis in COVID-19 (DOI: 10.1055 / a-1346-3178).
  3. Are the authors confident that their dates for enrolling patients in the study are correct? On January 1, 2020, the SARS-CoV-2 genome was not known yet.
  4. Figure 1 is of poor quality.
  5. Table 1 - the authors sometimes use full names, sometimes abbreviations. Would you mind systematizing it?
  6. Please also correct minor spelling mistakes (e.g., COVID-19, not Covid-19).
  7. The discussion could be extended to the latest observations on the pathomechanisms of thrombosis and bleeding in COVID-19.

Author Response

Reviewer 2:

US authors’ manuscript on the relationship between thrombotic and bleeding complications and severe COVID-19. The manuscript is prepared carefully. However, authors should clarify what is new in their research compared to previous works in this field.

Response: We thank the reviewer for their comments. In our discussion section, we have attempted to provide a comprehensive overview of what is already known (para 2, para 4, para 5) the results from randomized trials on this topic- some of which were contradictory (para 5) and what the present study add (para 1 and 3) while acknowledging the limitations and future directions (para 6). 

Please find below my comments. I am asking the authors to include them in the next version of the manuscript:

  1. Would you please add the name of the country to your affiliation?

Response: Definitely, it has been added

  1. In the introduction, the authors should also refer to the disorders of fibrinolysis in COVID-19 (DOI: 10.1055 / a-1346-3178).

Response: Thank you, this has been added – reference 4.

  1. Are the authors confident that their dates for enrolling patients in the study are correct? On January 1, 2020, the SARS-CoV-2 genome was not known yet.

Response: Thank you for the question. January 1, 2020 was just the start date for screening the patients for eligibility for the study. Agree that the SARS-COV-2 genome was not known by then. The first patient entered the study only later.

  1. Figure 1 is of poor quality.

Response: Added

  1. Table 1 - the authors sometimes use full names, sometimes abbreviations. Would you mind systematizing it?

Response: We used abbreviations for long terms that were previously used in the manuscripts along with expanded version of the abbreviations. Totally understand that this could be confusing. Therefore, we have added additional keywords to the table and provided explanation for the abbreviations used.

  1. Please also correct minor spelling mistakes (e.g., COVID-19, not Covid-19).

Response: Reviewed and edited

  1. The discussion could be extended to the latest observations on the pathomechanisms of thrombosis and bleeding in COVID-19.

Response: The findings from the most recent randomized trials on this topic and possible mechanisms has been included (para 2, para 5). There are several unknowns which would hopefully be addressed in future studies. It was helpful to review study mentioned by the reviewer (disorders of fibrinolysis in COVID-19 ,DOI: 10.1055 / a-1346-3178) and this has been added (reference 4). Thank you!

Round 2

Reviewer 1 Report

The authors answered my questions.

Reviewer 2 Report

The authors have addressed all of my concerns with the original manuscript. The revised manuscript is ready for publication.